# DECODING AS DYNAMIC PROGRAMMING FOR RECURRENT AUTOREGRESSIVE MODELS

**Najam Zaidi**
Faculty of Information Technology
Monash University, Australia
syed.zaidi1@monash.edu

**Trevor Cohn**
School of Computing and Information Systems
University of Melbourne, Australia
t.cohn@unimelb.edu.au

**Gholamreza Haffari**
Faculty of Information Technology
Monash University, Australia
gholamreza.haffari@monash.edu

## ABSTRACT

Decoding in autoregressive models (ARMs) consists of searching for a high scoring output sequence under the trained model. Standard decoding methods, based on unidirectional greedy algorithm or beam search, are suboptimal due to error propagation and myopic decisions which do not account for future steps in the generation process. In this paper we present a novel decoding approach based on the method of auxiliary coordinates (Carreira-Perpinan & Wang, 2014) to address the aforementioned shortcomings. Our method introduces discrete variables for output tokens, and auxiliary continuous variables representing the states of the underlying ARM. The auxiliary variables lead to a factor graph approximation of the ARM, whose maximum a posteriori (MAP) solution is found exactly using dynamic programming. The MAP solution is then used to recreate an improved factor graph approximation of the ARM via updated auxiliary variables. We then extend our approach to decode in an ensemble of ARMs, possibly with different generation orders, which is out of reach for the standard unidirectional decoding algorithms. Experiments on the text infilling task over SWAG and Daily Dialogue datasets show that our decoding method is superior to strong competing decoding methods.

## 1 INTRODUCTION

Neural autoregressive models (ARMs) have shown remarkable performance on various natural language processing tasks such as question answering, machine translation, summarization and reading comprehension (Anderson et al., 2018; Bahdanau et al., 2014; Wan et al., 2019). These models are usually trained in an end-to-end manner by optimizing the training objective to learn model parameters. Once the model has been trained, the output is generated by searching for a high scoring sequence given the context and the trained model. This is referred to as the decoding problem.

ARMs create a sequence by repeatedly generating the next symbol conditioned on *all* previous symbols generated. Symbols play dual duty: first they are generated, and next they are incorporated into the conditioning of subsequent decisions. As subsequent decisions can be arbitrarily distant in the sequence, these generation models are non-Markovian, i.e. they do not have a *bounded* Markov order. As prominent examples, ARMs include Elman's recurrent neural networks (RNNs) (Elman, 1990), conditional text generation models with RNN-based decoder (Sutskever et al., 2014), and transformers (Vaswani et al., 2017).

Exact decoding in ARMs is computationally hard, as the output search space is exponentially large and does not lend itself to efficient algorithms. This is due to *non-decomposable* long-range interdependencies among the output variables, i.e. an output token *directly* depends on all of the previously generated tokens. Standard uni-directional decoding algorithms, e.g. greedy and beam search,

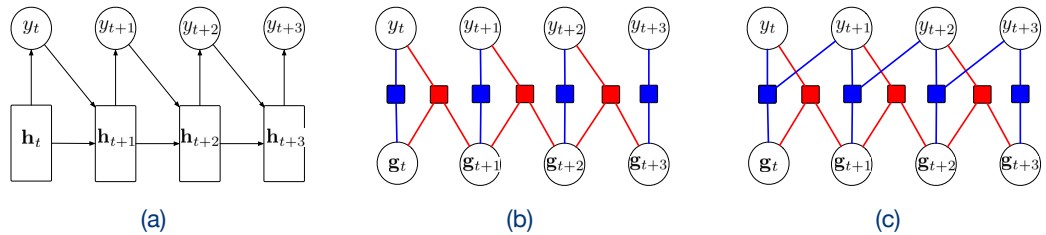

Figure 1: A typical RNN with unbounded Markov order is shown in (a). The factor graphs of our zero-order and first-order Markov approximations are illustrated in (b) and (c), respectively. The blue and red factors correspond to likelihood terms and the constraint violations, respectively, from equations (3), for order $k = 0$, and (4), for $k = 1$.

are ineffective in producing high-scoring output sequences, as errors in the decoding history can adversely affect the future. These algorithms make local decisions to extend an incomplete sequence (hypothesis) by selecting the token with the maximum likelihood at each time step, hoping to get a globally optimal complete sequence (Bahdanau et al., 2014; Sutskever et al., 2014; Mikolov et al., 2010).

In this paper, we present a novel decoding method, based on the method of auxiliary coordinate (MAC), which has been mainly investigated for training deep neural networks (Carreira-Perpinan & Wang, 2014). Our approach introduces discrete variables for output tokens, and *auxiliary* continuous variables representing the states of the underlying ARM. The auxiliary variables lead to a factor graph approximation of the ARM with a bounded Markov order (see Figure 1). We then alternate between optimizing over the output variables and the state variables. The state variables are updated to respect the state dynamics of the underlying ARM and the currently fixed output tokens. The output variables are updated by dynamic programming to exactly optimise a global scoring function, decomposed over local factors determined by the currently fixed state variables. We then extend our MAC-based decoding approach to decode under product of ARM experts (Hinton, 1999), i.e. an *ensemble* of ARMs combined additively in log-space, with each using a different generation order.

To validate our approach, we evaluate on the text infilling task, which consists of filling missing parts of a sentence or a paragraph (Horvat & Byrne, 2014; Tromble & Eisner, 2009; Schmaltz et al., 2016). Text infilling is challenging as it requires the global structure of the sentence to fill a blank properly. As shown in the figure to right, both greedy and beam search fail to fill the blanks correctly. We conduct ex-

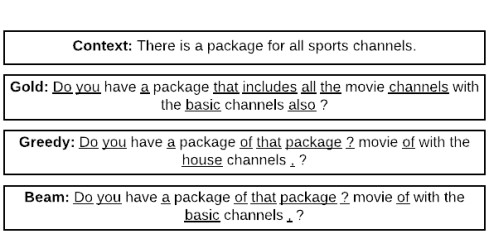

periments on two datasets: SWAG (Zellers et al., 2018) and Daily dialogue (Li et al., 2017) with various mask rates. We show that our decoding approach achieve remarkable improvements against the greedy and beam search algorithms as well as TIGS (Liu et al., 2019), a recently introduced strong inference method for this task. TIGS decodes by iteratively optimizing the output variables in a continuous relaxation of the discrete output space, and then projecting back the fractional solution to the discrete space. In contrast, the outputs in our approach stay in the discrete space, and we iteratively optimize over both the outputs and the continuous state variables.

## 2   RELATED WORK

**Inference in Relaxed Output Spaces.**   One approach to inference in deep neural networks is to relax the discrete output space to a continuous space, and then use gradient descent optimisation over the relaxed output variables, including structured prediction energy networks (SPENs) (Belanger & McCallum, 2016) and deep value networks (DVNs) (Gygli et al., 2017). The gradient descent optimisation for inference in SPENs is replaced by an inference network in Tu & Gimpel (2018), in order to speed up the prediction process. In all of these works, the relaxed output should be projected back to the discrete space at the end of the inference in the relaxed space. This is usually done by

*rounding*, similar to rounding the fractional solution of a linear program relaxation of an integer linear program. Performing the best projection, in terms of the loss function of interest, is usually a hard problem in itself; therefore, the quality of the solution is often degraded in the projection step.

Our inference approach is similar to that of SPENs and DVNs in that it iteratively optimises and refines the predicted output. However, the output variables always stay in the discrete space in our approach, so we do not need a projection step of the inference process for rounding. Our approach, instead, relaxes the space of continuous state variables to those which may not strictly satisfy the constraints imposed by the dynamics of the underlying ARMs. Cho (2016) proposed to inject *random* noise to the hidden states of a conditional recurrent neural language model during greedy or beam search, and execute multiple parallel decoding runs. Our approach, instead, makes structured changes to the state variables in the direction of optimising the decoding objective.

**Method of Auxiliary Coordinates.** Carreira-Perpinan & Wang (2014) introduced the method of auxiliary variables (MAC) to *train* a neural network. Their idea was to break the interdependence in layers of the neural network by introducing additional variables per data point and per hidden unit. This resulted in smaller problems that could be solved easily given a careful design of the architecture. Taylor et al. (2016) extends MAC by introducing different auxiliary variables for the linear operations and non-operations in a neural network. This results in three sets of variables involving two auxiliary variables and the weights, which is then optrimized by alternating direction method of multipliers (ADMM). Similarly, (Wang et al., 2019) introduces new variables and train the neural network using ADMM to globally optimize the training objective, given a careful design of the architecture.

## 3   Our Decoding Framework

**Notations.** We denote scalars, vectors and matrices using lower-case, bold lower-case and bold upper-case letters, e.g. $y$, $\mathbf{y}$ and $\mathbf{Y}$. Individual elements of $\mathbf{y}$ are denoted as $y_i$.

**Problem Formulation.** Consider a recurrent neural network model for text generation. The probability of generating a sequence $y_1, .., y_n$ under the RNN is decomposed as

$$P(y_1, .., y_n) = \prod_{i=1}^{n} P(y_i | \mathbf{y}_{<i}), \tag{1}$$

where $P(.|\mathbf{y}_{<i}) := \mathrm{softmax}(\mathbf{W}\mathbf{h}_i + \boldsymbol{b})$, and the state dynamics is $\mathbf{h}_i = \boldsymbol{f}(\mathbf{h}_{i-1}, y_{i-1})$. Decoding then refers to the following optimization problem,

$$\arg \max_{y_1, .., y_n} \log P(y_1, .., y_n) = \sum_{i=1}^{n} \log P(y_i | \mathbf{y}_{<i}) \tag{2}$$

The above optimization problem is computationally hard as it decomposes to conditional probabilities with unbounded length for the conditioning contexts, i.e. a non-Markovian model.

**Method of Auxiliary Coordinate (MAC).** Our goal is to decompose the optimisation problem in eqn (2) into smaller optimisation problems, which can be solved jointly and coupled via the state variables. This would make decoding more resilient compared t uni-directional decoding, where early errors can adversely affect the future.

Let us start by considering the following decomposition of the decoding problem,

$$\arg \min_{\mathbf{y}_1^n, \mathbf{g}_1^n} \quad -\sum_i \log P(y_i | \mathbf{g}_i) \tag{3}$$
$$\text{s.t.}$$
$$\mathbf{g}_i = \boldsymbol{f}(\mathbf{g}_{i-1}, y_{i-1}) \quad \forall i \in [2, .., n].$$

Notice the appearance of the new explicit variables $\mathbf{g}$ in this optimisation formulation of the decoding problem, which mirror the role of $\mathbf{h}$ in the underlying RNN. Eliminating the $\mathbf{g}$ variables would make this optimisation problem exactly equivalent to the original decoding problem in eqn (2). The

**g** serve as continuous auxiliary coordinates, following the MAC technique (Carreira-Perpinan & Wang, 2014).

The objective function in eqn (3) links the state variable $\mathbf{g}_i$ to token variable $y_i$ generated at the time step $i$. This is a *zero-order* decomposition of the log-likelihood function in eqn (2), as the terms in the objective do not condition on the previous tokens. Interestingly, we can generalise to a $k^{th}$ *order* decomposition by linking the state variable $\mathbf{g}_{i-k}$ from $k$ steps in the past to the generation of the current token $y_i$. This is due to the fact that the current hidden state, responsible for generating the current token in RNNs, is a *deterministic* function of the past hidden state $\mathbf{g}_{i-k}$ and all of the $k$ generated tokens between that time step and the current time step $\mathbf{y}_{i-k}^{i-1}$. Hence, our $k^{\text{th}}$ order decoding optimisation problem is written as follows:

$$\arg\min_{\mathbf{y}_1^n, \mathbf{g}_1^n} \quad -\sum_i \log P(y_i | \boldsymbol{f}^{(k)}(\mathbf{g}_{i-k}, \mathbf{y}_{i-k}^{i-1})) \tag{4}$$
$$\text{s.t.}$$
$$\mathbf{g}_i = \boldsymbol{f}(\mathbf{g}_{i-1}, y_{i-1}) \quad \forall i \in [2, .., n],$$

where $\boldsymbol{f}^{(k)}(\mathbf{g}_{i-k}, \mathbf{y}_{i-k}^{i-1})$ denotes $k$ repeated applications of the RNN's state transition function $\boldsymbol{f}(.)$ to compute the current hidden state from the past state $\mathbf{g}_{i-k}$ and the tokens observed since then $\mathbf{y}_{i-k}^{i-1}$. For example with $k = 2$, $\boldsymbol{f}^{(2)}(\mathbf{g}_{i-2}, \mathbf{y}_{i-2}^{i-1}) = \boldsymbol{f}(\mathbf{W}\boldsymbol{f}(\mathbf{W}\mathbf{g}_{i-2}, \mathbf{y}_{i-2}), \mathbf{y}_{i-1})$. The variables $g_i$ and $y_i$ for indices $i \leq 0$ are assigned null values, i.e., 0, or a sentence start sentinel. Assuming the constraints are satisfied, this constrained optimisation problem is equivalent to the decoding problem in eqn (2). Otherwise, when the constraints are not met, the $k^{\text{th}}$ order objective results in a more accurate approximation to the original decoding problem compared to the first-order formulation in eqn (3). As $k$ is made larger, the reliance on **g** is reduced, as $f^{(k)}$ uses the RNN's state transition function directly. This will be important in early stages of optimisation, where **g** does not accurately reflect the RNN state dynamic. The cost of using a higher order, is a higher complexity of inference, which we discuss in Section 4.

In summary, our decoding approach results in a constrained optimisation problem, incorporating two types of *factors* corresponding to the likelihood terms in the objective function and the constraint violations, respectively, which are denoted by blue and red, in the factor graph of Figure 1.

## 4 OPTIMISATION ALGORITHMS

We now turn to solving the constrained optimisation problem in eqn (4). Using the quadratic-penalty (QP) method (Nocedal & Wright, 2006), we turn it to an unconstrained optimisation problem,

$$\min_{\mathbf{g}_1^n, \mathbf{y}_1^n} \mathcal{L}(\mathbf{g}_1^n, \mathbf{y}_1^n, \mu) := \sum_i -\log P(y_i | \boldsymbol{f}^{(k)}(\mathbf{g}_{i-k}, \mathbf{y}_{i-k}^{i-1})) + \mu \|\mathbf{g}_i - \boldsymbol{f}(\mathbf{g}_{i-1}, y_{i-1})\|_2^2. \tag{5}$$

This suggests a two-step block coordinate descent algorithm to alternate between (i) optimizing **y**'s while **g**'s are fixed, and (ii) optimizing **g**'s while **y**'s are fixed. Other methods for constrained optimization can be used, e.g. the augmented Lagrangian, rather than the quadratic-penalty method (Nocedal & Wright, 2006; Taylor et al., 2016; Wang et al., 2019). However, the focus of our work is to investigate the effectiveness of decomposing the non-Markovian decoding objective to the bounded-Markov constrained optimisation problem in eqn (4), so we leave it to the future work to investigate the effect of different optimisation strategies.

It is worth noting that the above optimisation problem can be also interpreted as inference objective in an *stochastic* RNN, where the next hidden state $\mathbf{g}_i$ is conditionally generated based on the previous hidden state according to a multivariate Gaussian distribution with the mean vector $\mathbf{g}_{i-1}$ and the diagonal covariance matrix whose diagonal elements are $\mu^{-1}$.

**Updating the Output Variables** Assuming **g**'s are fixed, the discrete output variables **y**'s can be updated to *exactly* solve eqn (5) using a variant of the Viterbi algorithm (Viterbi, 1967). Let $P_{\mathbf{g}_{i-k}}(.|\mathbf{y}_{i-k}^{i-1}) := \text{softmax}(\mathbf{W} \cdot \boldsymbol{f}^{(k)}(\mathbf{g}_{i-k}, \mathbf{y}_{i-k}^{i-1}) + \boldsymbol{b})$ be the conditional probability of the next token given the previous $k$ tokens and $\mathbf{g}_{i-k}$. Consider a dynamic programming table $T \in R^{|\mathcal{Y}^k| \times n}$.

The table is filled from left to right, and at each time $i$, the element corresponding to the DP state $\mathbf{y}_{i-k+1}^i \in \mathcal{Y}^k$ is computed as,

$$T[\mathbf{y}_{i-k+1}^i, i] \leftarrow \min_{y' \in \mathcal{Y}} T[y' \circ \mathbf{y}_{i-k+1}^{i-1}, i-1] - \log P_{\mathbf{g}_{i-k}}(y_i | y' \circ \mathbf{y}_{i-k+1}^{i-1}) + \mu \|\mathbf{g}_i - \boldsymbol{f}(\mathbf{g}_{i-1}, y_{i-1})\|_2^2$$

where $y' \circ \mathbf{y}_{i-k+1}^{i-1}$ denotes the concatenation of the token $y'$ to the beginning of the sequence of tokens $\mathbf{y}_{i-k+1}^{i-1}$, which results in a sequence of tokens of length $k$. Once the DP table is built, the optimal sequence can be read off by traversing the table from the end towards the beginning. The time complexity of the DP is $O(n|\mathcal{Y}|^{k+1})$ where $n$ is the length of the sequence, $k$ is the Markov order, and $|\mathcal{Y}|$ is the size of the vocabulary.

**Updating the State Variables**    Next, we turn to optimizing the penalized decoding objective (eqn 5) with respect to the continuous auxiliary variables, $\mathbf{g}$, assuming fixed outputs, $\mathbf{y}$. For particular combinations of state transition and the likelihood functions, the optimal state values may have gradient-free closed form solution. However, we assume the general case, where the state transition and likelihood functions are given typical nonlinear functions. In the absence of closed-form solution, the simplest method is to use gradient-based optimization algorithms. Interestingly, the computational graph corresponding to the penalized decoding objective considers the state variables of *all* positions as the input when computing the output $\mathcal{L}(\mathbf{g}_1^n, \mathbf{y}_1^n, \mu)$. This means, there is no need for backpropagation through the time (BPTT), as opposed to the underlying RNN, to compute the gradient as the contribution of all positions is taken into account in parallel.

We also propose an approximation method to simplify the $\mathbf{g}$ update, by ignoring the log-likelihood term in the objective, which we denote by *forced decoding*. In this case, the penalty term can be optimized exactly, and reduced to zero by computing $\mathbf{g}$'s according to the RNN state transition function $\boldsymbol{f}(.)$ from-left-to-right while the output variables are fixed. This update does not involve computing the gradients, and only involves a forward pass through the RNN.

## 5    DECODING IN AN ENSEMBLE

In this section, we extend our MAC-based decoding approach to decode under an ensemble of ARMs, each using a different generation order. For example, consider two RNN-based language models, left-to-right (L2R) and right-to-left (R2L), where each of which gives a score to an assignment of words to the blank positions in the text infilling task. We are then interested to find an assignment of words which has the maximum sum of the scores under these two models. As both models are in the exponential family, this corresponds to a product of experts (Hinton, 1999). Unidirectional decoding algorithms cannot decode under such ensembles.

Let us consider the decoding problem in an ensemble of left-to-right and right-to-left RNNs,

$$\arg\max_{y_1, .., y_n} \sum_i \log \overrightarrow{P}(y_i | \mathbf{y}_{<i}) + \log \overleftarrow{P}(y_i | \mathbf{y}_{>i})$$

where each RNN has its own parameters. We then reformulate this optimisation problem, using auxiliary variables $\mathbf{g}_i$ and $\mathbf{g}_i'$ for the L2R and R2L RNNs, as follows,

$$\arg\min_{\mathbf{y}_1^n, \mathbf{g}_1^n, \mathbf{y}_1'^n, \mathbf{g}_1'^n} -\sum_i \log \overrightarrow{P}(y_i | \overrightarrow{\boldsymbol{f}^{(k)}}(\mathbf{g}_{i-k}, \mathbf{y}_{i-k}^{i-1})) + \log \overleftarrow{P}(y_i' | \overleftarrow{\boldsymbol{f}^{(k)}}(\mathbf{g}_{i+k}', \mathbf{y}_{i+k}'^{i+1}))  \tag{6}$$

$$\text{s.t.}$$
$$\mathbf{g}_i = \overrightarrow{\boldsymbol{f}}(\mathbf{g}_{i-1}, y_{i-1})  \quad \forall i \in [2, .., n]$$
$$\mathbf{g}_i' = \overleftarrow{\boldsymbol{f}}(\mathbf{g}_{i+1}', y_{i+1})  \quad \forall i \in [1, .., n-1]$$
$$\boldsymbol{e}[y_i] = \boldsymbol{e}[y_i']  \quad\quad\quad \forall i \in [1, .., n]$$

where the constraints in eqn (6) couple the two optimisation problems corresponding to the L2R and R2L RNNs. Note that we have enforced the equality between the *embeddings* of the words $y_i$ and $y_i'$ produced by the L2R and R2L models, respectively (denoted $\boldsymbol{e}[y]$). This provides a denser signal, e.g. from synonyms or words with related syntactic categories, in order to couple the two optimisation problems, compared to a sparse signal from constraints using *identity* of the tokens.

| Context: Look , this one matches our room and it's inexpensive .
Target: Moreover , it's easy to clean , right ? You are really lazy . |
| :--- |
| Context: Good evening , sir . Are you Mr . Jim Stewart from the States ?
Target: Ah , yes , that's right . |
| Context: So , what can I do for you today ? Are you needing to withdraw or
transfer ?
Target: I'm going to need a Deposit Certification , to handle the affairs
related to home . |

| Context: someone leans down , placing his head on his mother 's shoulder .
Target: a soldier is manning a gun from inside the helicopter . |
| :--- |
| Context: she looks off in another direction , slightly behind the office , and sees .
Target: a path from the motel office leads directly up to this house . |
| Context: as the detective descends , someone reaches into his pocket and pulls
out a pack of gum .
Target: he holds it out to someone . |

(a)  (b)

Figure 2: Some test examples from (a) Daily Dialogue and (b) SWAG datasets.

To solve the above optimisation problem, we use of the quadratic penalty method, similar to the previous section, i.e.,

$$\min_{\mathbf{g}_1^n, \mathbf{y}_1^n, \mathbf{g}'^n_1, \mathbf{y}'^n_1} \sum_i -\log \overrightarrow{P}(y_i | \overrightarrow{\boldsymbol{f}^{(k)}}(\mathbf{g}_{i-k}, \mathbf{y}_{i-k}^{i-1})) - \log \overleftarrow{P}(y_i' | \overleftarrow{\boldsymbol{f}^{(k)}}(\mathbf{g}'_{i+k}, \mathbf{y}'^{i+1}_{i+k}))$$

$$+\mu || \mathbf{g}_i - \overrightarrow{\boldsymbol{f}}(\mathbf{g}_{i-1}, y_{i-1})||_2^2 + \mu' || \mathbf{g}'_i - \overleftarrow{\boldsymbol{f}}(\mathbf{g}'_{i+1}, y'_{i+1})||_2^2 + \mu'' || \boldsymbol{e}[y_i] - \boldsymbol{e}[y_i']||^2 .$$

This suggests an optimisation algorithm which alternates between (i) updating $\{\mathbf{y}_1^n, \mathbf{g}_1^n\}$ in the first phase while the other variables are fixed, and (ii) updating $\{\mathbf{y}'^n_1, \mathbf{g}'^n_1\}$ in the second phase while the other variables are fixed. The updates for each of these phases is done using an iterative algorithm, similar to those presented in section 4, for updating the output and state variables. The only modification in the objective function of each phase (compared to the previous section) is the inclusion of the token embedding constraints $||\boldsymbol{e}[y_i] - \boldsymbol{e}[y_i']||^2$, which can be easily accounted for in the dynamic programming algorithm when updating the output variables.

## 6 EXPERIMENTS

**The Text Infilling Task.** Text infilling consists of predicting missing parts of a sentence or a paragraph. The task is encountered in everyday applications of restoring historical or damaged documents (Zhu et al., 2019), writing articles or contracts with templates (Ippolito et al., 2019) and text editing (Feng et al., 2019). Although important, text infilling is less explored and has been studied under simplified and restricted settings (Fedus et al., 2018; Zweig & Burges, 2011; Holtzman et al., 2018; Fan et al., 2018). For example, Horvat & Byrne (2014) restricted the vocabulary to the set of gold standard words blanked in the sentence. Here we follow a setup similar to (Liu et al., 2019), which also limits the vocabulary, but places no restriction on the number and position of the blanks, and thus is more similar to situations encountered in real life applications.

More formally let $B$ be a *mask*, comprising set of indices where blanks appear in the sentence. Let $\mathbf{y}^B$ be a target sentence where tokens at the positions of the sentence in $B$ have been masked. For example if $B = \{i, i + 1\}$ then $\mathbf{y}^B$ is $\{y_1, ...y_{i-1}, \_, \_, y_{i+2}, ..y_n\}$. Given the context $\mathbf{x}$ and masked sentence $\mathbf{y}^B$, the aim is to fill in the blanks as they appear in sentence. This requires considering global structure of the sentence along with the conditioning context.

**Datasets.** We evaluate our proposed approach on two text infilling tasks over two widely used publicly available corpora. The first task is conversation reply with a template (denoted as Daily) which is conducted on the DailyDialog dataset (Li et al., 2017). Similar to Liu et al. (2019) we convert the multi-turn dialogues into single-turn dialogues , resulting in 82,372 conversation pairs. The query sentence is used as input to the encoder $\mathbf{x}$ where as the reply is the output $\mathbf{y}$ from the decoder. The second task is captions from movies along with an ending (denoted as SWAG) which is conducted on the SWAG dataset (Zellers et al., 2018). We only consider the correct endings to build the target side. This gives us 73,000 pairs of sentences. The input to the encoder is the caption $\mathbf{x}$ whereas the decoder has to produce the ending $\mathbf{y}$ conditioned on $\mathbf{x}$. Some examples of these datasets are shown in Figure 2.

**Training.** We trained both left-to-right (L2R ) and right-to-left (R2L ) models, where R2L models are trained by reversing the target side sentence. All models are trained with a word embedding

|  | Decoding Method | Dialogue | | SWAG | |
|---|---|---|---|---|---|
|  |  | BLEU | PPLX | BLEU | PPLX |
| L2R | Greedy | 71.2 | 5.88 | 71.6 | 5.64 |
| | Beam | 71.3 | 5.84 | 71.8 | 5.58 |
| | TIGS | 73.0 | 4.31 | 74.0 | 3.49 |
| | Ours | **79.3** | **3.49** | **83.9** | **2.31** |
| R2L | Greedy | 70.2 | 6.96 | 63.9 | 7.07 |
| | Beam | 70.4 | 6.90 | 64.1 | 6.97 |
| | TIGS | 71.8 | 4.87 | 66.5 | 4.68 |
| | Ours | **77.9** | **3.80** | **78.7** | **3.04** |
| Both | Ours | 80.2 | - | 79.3 | - |
| | L2R component | - | 3.30 | - | 3.39 |
| | R2L component | - | 3.73 | - | 4.15 |

Table 1: BLEU score and perplexity of various models on the two datasets with 50% masking rate. The results of our decoding approach is based on the 1st order approximation.

size and hidden dimension size of 512. We use ADAM optimiser to train the models with an initial learning rate of 0.001. Since the source and target sides are in the same language, we shared the word embeddings between the encoder and decoder. The models were trained for 10 epochs.

**Baselines.** Our baselines include: greedy decoding, beam search, and a strong recently proposed inference algorithm for the text infilling task TIGS (Liu et al., 2019). TIGS decodes by iterating through the following steps: (i) relaxing the space of output variables from discrete to continuous and optimise over the continuous output variables using gradient based optimization, and (ii) projecting back the solution from the continuous space to discrete. In contrast, the outputs in our approach stay in the discrete space, and we optimize over both discrete and the continuous state variables, as part of an iterative coordinate descent procedure. Our methods and the baslines are implemented on top of OpenNMT[1] (Klein et al., 2017).

**Decoding Parameters** We use the Nesterov optimiser with a learning rate of 0.1. We experiment with Adam and simple SGD and find empirically that Nesterov works better than the other optimisers. Nesterov is a momentum based optimiser that stabilize the update directions and seems to better escape from poor local optima during the decoding iterations. Rather than taking a step in the direction of updated accumulated gradient, Nesterov optimiser first moves in the direction of previously accumulated gradient. It calculates the new gradient and then makes a correction. This prevents the optimiser from going too fast and results in increased responsiveness, which significantly improves the performance.

All $\mu$'s for the penalty terms corresponding to different constraints, are initialised with 0.5 and are multiplied by 1.2 after 5 iterations, and decoding was run for 10 iterations, chosen as the first order method had reliably converged in terms of objective value and the output string.

We use the RNN states **h** corresponding to the beam search solution to initialize the **g** variables in our decoding method. We experiment with initialising the hidden states with random values, or using the values corresponding to the greedy solution. We find that using beam search for initialisation gives better results. Compared to beam search, random initialisation requires an average of 12x more iterations, and would occasionally suffer from non-convergence, where the solution oscillates. We find negligible difference between the number of iterations needed for convergence between the beam and greedy search initialised hidden states.

## 6.1 RESULTS

The results are reported in Table 1. Following Liu et al. (2019), we build a test set of 5000 sentences for each dataset. We use a 50% masking rate and randomly place the blanks for each sentence. We perform experiments on left-to-right (L2R), right-to-left (R2L), and an ensemble of the two.

---

[1]https://github.com/Najamxaidi/Decoding-as-a-dynamic-program-for-recurrent-autoregressive-models.git

Generally, L2R models outperform R2L models. This may be due to sentences being generated inherently in a left to right manner. Hence modelling the writing process with a right to left model may make it difficult to learn useful patterns. The trend across all the models and both datasets is that a decrease in perplexity leads to a better BLEU score. The benchmark TIGS method (Liu et al., 2019) outperforms both the greedy algorithm and beam search. However, our decoding method outperforms TIGS and other baselines significantly. Compared to the greedy algorithm and beam search, our method and TIGS leverage information from both future and the past. Compared to TIGS, our method operates by keeping the output variables in the discrete space, whereas in TIGS the output variables are relaxed to the continuous space and projected back to the discrete space.

Given that context from both directions helps in better decoding when working with unidirectional models, we perform experiments on an ensemble of L2R and R2L models. Notably, the unidirectional greedy algorithm and beam search cannot operate on the ensemble. Our method, instead, can decode with the ensemble, which further improves the BLEU score for the Dialogue task.

## 6.2 ANALYSIS

**Varying the Masking Rate.** Increasing the masking rate makes it difficult for all the models to correctly fill in the blanks. We experiment on the dialogue dataset by randomly masking the test set with the rates 25%, 50%, and 75%. Results are reported in Table 2. As the masking rate increases, BLEU score decreases, whereas perplexity increases. Compared to the other techniques, our method is able to achieve better results even with high masking rates.

| Mask Rate | 25% | | 50% | | 75% | |
|---|---|---|---|---|---|---|
| | BLEU | PPLX | BLEU | PPLX | BLEU | PPLX |
| Greedy | 85.4 | 4.43 | 71.2 | 5.88 | 60.4 | 4.25 |
| Beam | 85.4 | 4.43 | 71.3 | 5.84 | 62.0 | 4.03 |
| TIGS | 88.0 | 3.47 | 73.0 | 4.31 | 62.5 | 3.53 |
| Ours (1st order) | **90.9** | **2.80** | **79.3** | **3.49** | **64.3** | **2.47** |

Table 2: Performance with varying masking rates for the different decoding methods.

**Varying the Markov Order.** Our *penalized* decoding objective is composed of the negative log-likelihood term and the penalty (see eqn 5). As the algorithm proceeds, both of these terms decrease, showing consistency in the auxiliary variables. We hypothesise that, although higher order Markov models produce more accurate approximations to the original decoding problem, they result in *harder* optimization problems. The following figure (right) plots the penalized decoding objective in $k$-th order Markov approximations for $k \in \{0, 1, 2\}$. The results are based on 100 examples in the test set for the Daily Dialogue dataset, with random masking using 50% masking rate. Observe that the penalized decoding objective tends to decrease less for the 2nd order method compared to the others. Table 3 reports the perplexity, BLEU score, and decoding time for these different Markov approx-

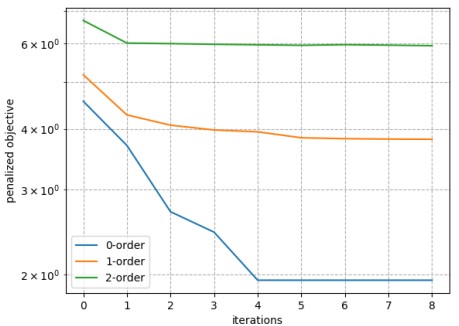

Figure 3: The figure shows the penalised decoding objective for zero, first and second order Markov models

imations. The results confirm the increase in the BLEU score and decrease in the perplexity, as the Markov order increases. Furthermore, it shows the trade-off in the solution quality vs. decoding time for these different approximations. In Table 3, we report results based on two different methods for updating the state variables in our method, i.e. gradient-based and forced decoding. As expected, gradient-based updates produced more accurate results compared to forced decoding, but at the cost of a longer run-time.

Figure 4 shows the iterative improvement of an example test sentence in each iteration of our decoding methods with varying Markov order. Each method is given the same initial solution produced by the beam search. The zero-order method converges to a bad solution in the first iteration and gets

|  | Gradient Based | | | Forced Decoding | | |
|---|---|---|---|---|---|---|
|  | BLEU | PPLX | Time | BLEU | PPLX | Time |
| 0-order | 77.4 | 6.55 | 297s | 75.4 | 6.86 | 65s |
| 1-order | 84.2 | 3.42 | 348s | 82.6 | 3.75 | 93s |
| 2-order | 85.5 | 2.33 | 893s | 83.4 | 3.34 | 711s |

Table 3: The results of varying Markov order and state variable update method. Time is reported for processing 100 sentences.

stuck there in future iterations. The first-order method arrives to a further improved solution while the second order method can find the best solution among all.

Figure 4: Improvement of the sentence in different iterations

## 7 CONCLUSION

This work presented a method for improving decoding in discrete autoregressive models using dynamic programming. The core idea is to introduce auxiliary variables to decouple the non-Markovian aspects of the model, permitting an approximate solution. This solution is used to create the next model approximation, and the process iterates. Our results show that our decoding framework is effective, leading to substantial improvements over greedy and beam search baselines. Our approach does have limitations, most notably the computational complexity which is polynomial in the vocabulary size, thus limiting its application to open text generation problems. Improving the complexity of decoding is an important direction for future research, as is applying the method to other autoregressive models, such as the Transformer, which includes self attention, as well as other structured prediction problems.

## ACKNOWLEDGMENTS

This work was supported by the Australian Research Council (DP160102686) and Multi-modal Australian ScienceS Imaging and Visualisation Environment (MASSIVE) (www.massive.org.au) through computational infrastructure. The authors are grateful to the anonymous reviewers for their insightful comments and corrections.

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
