# OpenReview forum: "Decoding As Dynamic Programming For Recurrent Autoregressive Models"
_ICLR.cc/2020/Conference — Accept (Poster)_

### Official Review · AnonReviewer3 · 2019-10-22
**Official Blind Review #3**

**Rating:** 8

**Review:**

I think this is a good study, unless I miss something. It proposes a new solution to one of the fundamental problems in the field, which significantly improves previous solutions in terms of accuracy. I will recommend it for acceptance unless I miss something. (I’m not an expert in the field and the problem seems to be so fundamental, making me cautious about judging the novelty of the proposed solution.)

There seems to be some unclarity about the optimization algorithm. In short, I suspect that the proposed optimization has some difficulty with its convergence.
-	In Sec.4, the authors suggest “a two-step block coordinate descent algorithm to alternate between (i) optimizing y’s while g’s are fixed, and (ii) optimizing g’s while y’s are fixed”. They further suggest `a variant of the Viterbi algorithm’ for (i) and gradient-based algorithms for (ii). These seem to make sense to me so far.
-	Then, in Sec.6, they state that “We use the Nesterov optimizer with a learning rate of 0.1. All \mu’s are initialised with 0.5 and are multiplied by 1.2 after 5 epochs”. I think this needs some explanation.
-	For instance, what is the thought behind the choice of the optimizer? Why was Nesterov’s acceleration necessary instead of plain GD? Is it essential to use the scheduled adjustment of \mu?
-	It states “All \mu’s” but there seems only a single \mu in Eq.(5). Am I missing something?
-	 Some explanation on what “Nesterov optimizer” and \mu would also be necessary for a wide range of readers.
-	There is a statement on the initialization of the optimization in p.7: “h corresponding to the beam search solution to initialize the g variables”. How sensitive is the optimization to the initial values? For instance, will the results change if g’s are initialized by the solution of the greedy method.
-	There is a plot without a figure caption in p.8, which shows how the objective cost decreases with parameter updates. Why are their values at ‘epoch’=0 lower than those at the subsequent epochs? Does this mean the initial values give more optimal parameters?

Additional comments:
-	It would be more friendly to the readers to show the definition of the notations like y^n_1.
-	There is a statement like “Decoding was run for 10 epochs.” I suppose epochs here mean iteration count of the alternated parameter updates in the proposed algorithm. Why do the authors call it `epoch’?
-	No figure caption for the plot in p.8. It also uses `epochs’ for the horizontal axis title. No axis title for the vertical axis.

**Experience Assessment:**

I have read many papers in this area.

**Review Assessment: Checking Correctness Of Derivations And Theory:**

I assessed the sensibility of the derivations and theory.

**Review Assessment: Checking Correctness Of Experiments:**

I carefully checked the experiments.

**Review Assessment: Thoroughness In Paper Reading:**

I read the paper at least twice and used my best judgement in assessing the paper.

---

> ### Author Response · Authors · 2019-11-15
> **Reply to reviewer #3**
>
> We appreciate the reviewer’s constructive comments.
>
> Q. Novelty of the proposed solution.
>
> A. To the best of our knowledge, this is the first work to use the method of auxiliary variables in order to solve the decoding problem in ARMs. As discussed in our paper, the current approaches either use greedy/beam search or rely on techniques that involve relaxation of the discrete output space to continuous, followed by a projection step to convert the ‘fractional’ solution to an ‘integral’ solution . In contrast, our method does not require such a relaxation, hence does not need any projection step which may harm the solution. The experimental results show that our method outperforms not only popular decoding algorithm (i.e. greedy/beam search), but also TIGS which is a state-of-the-art inference method for the text infilling task. TIGS is a strong decoding method representing the inference approach of relaxing the discrete output space followed by projection.
>
> Q. What is the thought behind the choice of the optimizer? Why Nesterov’s instead of plain GD?
>
> A. The optimiser was chosen based upon experiments. We tried different optimisers including Adam, simple SGD and found the Nesterov optimiser worked the best. This momentum-based optimizer stabilizes update directions and seems to better escape from poor local optima during the decoding iterations.
>
> Q. Is it essential to use the scheduled adjustment of \mu?
>
> A. From the theoretical point of view, the theorems on the convergence proofs of the quadratic penalty for solving constrained optimisation problems (such as those resulted from our decoding approach) rely on increasing sequences of \mu (eg theorems 17.1 and 17.2 in [1]). Empirically, we found that increasing \mu by multiplying it with 1.2 every 5 iterations works well. Increasing \mu’s is also done in [2], which proposes the method of auxiliary coordinates for gradient-free training of feed forward neural networks.
>
> [1] J. Nocedal and S. J. Wright. Numerical Optimization. Springer Series in Operations Research and Financial Engineering. Springer-Verlag, New York, second edition, 2006.
> [2] Miguel Carreira-Perpinan and Weiran Wang. Distributed optimization of deeply nested systems. In Artificial Intelligence and Statistics (AISTATS), pp. 10–19, 2014
>
>
> Q. It states “All \mu’s” but there seems only a single \mu in Eq.(5).
>
> A.  Thanks, we meant different \mu’s for the penalty terms corresponding to different constraints. However, we have used one \mu for all constraints in this work, so we have updated the text.
>
> Q. Regarding initialisation of the g variables.
>
> A. We observe that beam initialisation is better than greedy, while random initialisation requires more steps then greedy or beam to reach convergence. Compared to beam, random initialisation requires an average of 12x more iterations. Sometimes the solution found using random initialisation is poor, and the solution oscillates without convergence.
>
> Q. There is a plot without a figure caption in p.8...Why are their values at ‘epoch’=0 lower than those at the subsequent epochs?
>
> A. We spotted a plotting error, which is now fixed. The new plot has been added together with the axis label and figure caption.
>
> Additional comments:
> We have put a paragraph on the notation in the updated paper.
> We have replaced the word ‘epoch’ with ‘iteration’ in the updated version of the paper.

---

### Official Review · AnonReviewer1 · 2019-10-22
**Official Blind Review #1**

**Rating:** 6

**Review:**

The paper proposes a decoding algorithm for auto-regressive models (ARMs) to improve the decoding accuracy. The key idea is to introduce auxiliary continuous states variables, and alternatively optimize the discrete output variables (i.e., the tokens) and the continuous states. Given the hidden states, the decoding can be efficiently done by veterbi-like algorithms. Then the constraint of the auxiliary variables can be imposed by penalty based continuous based optimization. The paper also extends the idea to the ensemble of ARMs.  On two NLP tasks, the paper shows improvement upon the existing greedy or beach search based approaches.

Overall, the proposed method can be very useful in RNN decoding and structure prediction. The idea, however, is quite straightforward to people who are familiar with probabilistic graphic models and inference. I am surprised (if the authors' claim is correct) the RNN community still relies on greedy/beam search for decoding. In graphical model language, the hidden states (h_i or g_i in the paper) are typically viewed as continuous latent random variables. The function f that links consecutive hidden states are factors or potential functions. For inference/prediction, you can jointly optimize the hidden states and discrete outputs, where the alternative updates are a natural choice. Similar ideas were used long time ago when (hierarhical) conditional random fields were popular. Honestly, I don't see anything new here.  Here are a few comments:

1. Don't say ARMs are non-Markov model and/or with unbounded Markov order. This is very misleading and over bragging. RNNs are just a nonlinear version of HMM/hidden Kalman filter. The only difference is that the states are continuous (Kalman filter uses continuous states as well), and the state transition kernel is nonlinear, constructed in a very black-box way. People like to make some analogy with brains --- unfortunately, these explanations are at most an analogy.  Given hidden states, RNNs are just first order Markov-chains, nothing special. If you integrate out hidden states, of course, every output is dependent on all the previous outputs. But the same argument applies to all the hidden markov models/dynamic systems. This is not something unique to RNNs, and shouldn't be hyped everywhere.

2. Is there a way to show the standard deviations/error bars in the test results, say, Table 1 & 2 and the figure? One single number is usually not representative for the performance, unless you have a large test set.

**Experience Assessment:**

I do not know much about this area.

**Review Assessment: Checking Correctness Of Derivations And Theory:**

I carefully checked the derivations and theory.

**Review Assessment: Checking Correctness Of Experiments:**

I assessed the sensibility of the experiments.

**Review Assessment: Thoroughness In Paper Reading:**

I read the paper thoroughly.

---

> ### Author Response · Authors · 2019-11-15
> **Reply to reviewer #1**
>
> We thank the reviewer for the insightful comments.
>
> Q. Don't say ARMs are non-Markov model and/or with unbounded Markov order…. RNNs are just a nonlinear version of HMM/hidden Kalman filter.
> A.  We agree that an RNN decoder can be considered as a non-linear HMM/hidden Kalman filter. However, we note that RNNs are a subclass of the more general class of ARMs which involve other architectures as well, such as Transformer decoders. Please note that we have mentioned the unbounded Markov property for the general class of ARMs.
>
> To elaborate further, one of the crucial differences between the Transformer and RNN decoders is the dependency of the next state to the previous states. In Transformer decoders, the next state is an immediate function of ‘all’ previous states, i.e. S_t = f(S_{t-1},..,S_1). However in vanilla RNN decoders, the next state is an immediate function of only the previous state, i.e. S_t = f(S_{t-1}). Therefore, Transformer decoders (as ARMs) do not have a bounded Markov order, as having only the previous state is not enough for computing the next state.
>
> Q. I am surprised (if the authors' claim is correct) the RNN community still relies on greedy/beam search for decoding.
> A. Greedy/Beam search are the most popular decoding algorithms in the RNN community (eg [1]). The NLP community, for example, use beam/greedy search algorithms prominently for various sequence-to-sequence tasks including machine translation (eg [2]) and summarisation (eg [3]), to name a few. Designing better decoding algorithms is an active research area, which includes this paper and the papers discussed in section 2.
>
> [1] E Cohen and J Beck, “Empirical Analysis of Beam Search Performance Degradation in Neural Sequence Models”, ICML, 2019.
> [2]  Gu, J., Wang, Y., Cho, K., & Li, V.O. “Improved Zero-shot Neural Machine Translation via Ignoring Spurious Correlations”, ACL, 2019.
> [3] Liu, Y., & Lapata, “M. Hierarchical Transformers for Multi-Document Summarization”, ACL, 2019.
>
> Q. The idea, however, is quite straightforward to people who are familiar with probabilistic graphical models and inference.
> A. One of the key contributions of the paper is to propose a decoding method based on the method of auxiliary variables and link the resulting approach to probabilistic graphical models (PGMs). Once the connection is made clear, the rest of the approach becomes clearer for researchers with a background in PGMs (although there are important details to figure out for alternate optimization of the hidden states and the output variables to make them efficient and effective).
>
> Note that our approach considers (i) high-order factors capturing the dependency of an output variable with its ‘several’ previous variables (similarly, high-order state dependencies can be considered), and (ii) an extension to inference for ensemble of ARMs (in our application, left-to-right and right-to-left models).
>
> To the best of our knowledge, our paper is the first to formulate the inference problem in ARMs by the method of auxiliary variables, and linking it to the factor graphs with high-order dependency factors as a way of approximating the underlying model. Furthermore, our paper presents solid experimental results showing that the resulting decoding method outperforms not only the most popular decoding algorithms (i.e. greedy and beam search), but also the state-of-the-art TIGS algorithm on the text infilling task.
>
> Q. The error bars.
>
> A.  Our test set contains 5000 sentence pairs which is quite large, and accordingly the error bars will be small. However, we will add the error bars to the tables and figures in the new version of the paper.

---

### Official Review · AnonReviewer2 · 2019-10-25
**Official Blind Review #2**

**Rating:** 6

**Review:**

This work introduces a new algorithm to improve decoding in discrete autoregressive models (ARMs). Because exact decoding of ARMs is computationally difficult, the contribution of this paper is to introduce a new approximate solution. The authors show experimentally that this new framework is very effective and outperforms multiple baselines such as greedy and bean search and another text filing algorithm called TIGC on a text-filling benchmark. This work is not in my domain of expertise so I am not able to have a very accurate evaluation. However, based on the references cited in this paper, there are other approximate solution for ARMs and I believe the authors need to use those as baselines to show that the proposed approximate solution is useful.

**Experience Assessment:**

I do not know much about this area.

**Review Assessment: Checking Correctness Of Derivations And Theory:**

I did not assess the derivations or theory.

**Review Assessment: Checking Correctness Of Experiments:**

I assessed the sensibility of the experiments.

**Review Assessment: Thoroughness In Paper Reading:**

I made a quick assessment of this paper.

---

> ### Author Response · Authors · 2019-11-15
> **Reply to reviewer #2**
>
> We thank the reviewer for the feedback.  We would like to make the following clarifications.
>
> Q. Comparison with other approximate inference methods for ARMs.
> A. In addition to comparison with classical inference methods, i.e. greedy and beam search, we have also compared our inference technique with TIGS, which is introduced in ACL 2019 [1]. TIGS is state-of-the-art inference method for the text infilling task. It is a strong and sophisticated inference method representative of the body of the literature which consider a continuous relaxation of the discrete optimisation involved in the decoding process. These methods then use algorithms for continuous optimisation to solve the relaxed problem, and then project back the ‘fractional’ solution to the discrete space (c.f. please see the Related Work section in the paper for more details).
>
> Given that our method outperforms TIGS in comparable experimental conditions to [1], it implies that our method will outperform the other inference methods which are compared with TIGS as well [1], including bidirectional beam search and inference algorithms that can be applied directly to generative models including [2,3].
>
> [1]  Dayiheng Liu, Jie Fu, Pengfei Liu, and Jiancheng Lv. Tigs: An inference algorithm for text infilling with gradient search. ACL, 2019
> [2] Berglund, M., Raiko, T., Honkala, M., Kärkkäinen, L., Vetek, A., & Karhunen, J. T. (2015). Bidirectional recurrent neural networks as generative models. In Advances in Neural Information Processing Systems (pp. 856-864).
> [3] Sun, Q., Lee, S., & Batra, D. (2017). Bidirectional beam search: Forward-backward inference in neural sequence models for fill-in-the-blank image captioning. In Proceedings of the IEEE Conference on Computer Vision and Pattern Recognition (pp. 6961-6969).

---

### Comment · Area_Chair1 · 2019-11-15
**Reviewers, any comments on author response?**

Dear Reviewers, thanks for your thoughtful input on this submission!  The authors have now responded to your comments.  Please be sure to go through their replies and revisions.  If you have additional feedback or questions, it would be great to know.  The authors still have one more day to respond/revise further.  Thanks!

---

### Decision · Program_Chairs · 2019-12-19

**Decision:**

Accept (Poster)

**Comment:**

This paper proposes an approximate inference approach for decoding in autoregressive models, based on the method of auxiliary coordinates,  which uses iterative factor graph approximations of the model.  The approach leads to nice improvements in performance on a text infilling task.  The reviewers were generally positive about this paper, though there was a concern that more baselines are needed and discussion was very limited following the author responses.  I tend to agree with the authors that their results are convincing on the infilling task.  The impact of the paper is a bit limited by the lack of experiments on more standard decoding tasks, which, as the authors point out, would be challenging as their approach is computationally demanding.  Overall I believe this would be an interesting contribution to the ICLR community.